# Relationships between Green Space Attendance, Perceived Crowdedness, Perceived Beauty and Prosocial Behavior in Time of Health Crisis

**DOI:** 10.3390/ijerph19116778

**Published:** 2022-06-01

**Authors:** Tania Noël, Benoit Dardenne

**Affiliations:** Psychology and Neuroscience of Cognition Research Unit, University of Liège, 4000 Liège, Belgium; b.dardenne@uliege.be

**Keywords:** urban green space, social orientation, prosocial behavior, crowdedness, COVID-19, pandemic

## Abstract

An emergent body of evidence shows the impact of exposure to nature on prosocial attitudes and interpersonal relationships. This study examines relationships between green space (GS) attendance, perceived beauty of the space, perceived crowdedness of the space, and prosocial behavior. A cross-sectional study with snowball sampling was conducted in April 2020. All participants (N = 1206) responded to an online survey that included a French version of the social value orientation slider measure (used as a proxy for prosocial behavior), questions about the lockdown, and their GS attendance. After retaining only participants who had visited a GS at least once since the beginning of their lockdown (N = 610), multiple linear regressions showed that social orientation scores demonstrated associations with the interaction between GS attendance and perceived crowdedness of the GS, suggesting that attending low crowded GS is linked to increasing prosociality. These results provide insight into the roles that GS can have during a health crisis and suggest some practical implications.

## 1. Introduction

Human society and cities suffer from various crises at any time, including pandemics like the H1N1 virus, polio, Ebola, or Zika, which we face in the current century. In early 2020, the COVID-19 pandemic was declared a public health emergency of international concern by the World Health Organization (WHO), WHO’s highest alarm level [1]. If pandemics have always existed, their occurrence keeps growing, and the explanation possibly lies in the environmental crisis we are currently experiencing. Studies show that the diversity of human pathogens among nations is positively associated with biodiversity, i.e., the diversity of wildlife species [2]. The loss of biodiversity in ecosystems creates the general conditions favoring and making possible disease pandemics such as COVID-19 [2,3,4]. A lot of factors, for example, deforestation or poorly regulated agricultural surfaces, contribute to altering the composition of wildlife communities, significantly increase the contact of humans with wildlife, alter niches that harbor pathogens, and increase their chance to come in contact with humans [3]. We probably have to be prepared for other pandemics in the next coming years. In the current context, it is important to reflect on how to improve our capacity to deal with such crises, while considering that solutions have to be sustainable, i.e., that they consider both human and environmental aspects. Taking into account the social impact of the envisaged solutions means also paying particular attention to lower-income groups as it now seems accepted that COVID-19 is hitting these groups the hardest [5]. In this study, we would like to highlight the potential of publicly accessible urban green spaces (GS) to help face the consequences of crises such as pandemics. Public urban GS consist mainly of semi-natural areas, referring in the present study to any vegetation found in the urban environment (i.e., towns, cities, suburbs, and their surroundings, which are characterized by high population density and built environment infrastructure) accessible to everyone without restriction (e.g., parks, playgrounds, walking paths, yards, plazas, peri-urban forests, road and rail networks, and their associated land, etc.). Focusing on public accessible nature seems important, given that we know that it is often the poorest groups who live without access to private gardens [6]. However, it is important to keep in mind that, currently, public accessible GS are not equally available to all population groups [7,8].

While previous studies seem to show that GS attendance is linked to an increase in prosociality, partially explained by the perception of the characteristics of the space itself [9], the present study takes place in the very particular context of a health crisis. Therefore, we not only focus on the link between GS attendance and prosociality by including perceived physical characteristics of the space as it has been designed or maintained (i.e., perceived beauty) but also by taking into account more ‘social’ characteristics of the space by including the perception of crowdedness. This variable has not yet been included in studies investigating the link between urban GS and prosocial behavior to the best of our knowledge. The perceived crowdedness of GS was, therefore, of particular interest in the present study.

### 1.1. Green Spaces as Resilience Infrastructures

It is already known that GS represent efficient resilience infrastructures, especially during pandemics. First, in “dense” cities (i.e., high number of people living and working in a certain area [10]), where high infection rates have often been reported [11], public open spaces like parks allow people to avoid crowding (i.e., how close everyone is to each other at a given time and place [10]) and, thus, decrease disease transmission rate [5]. Secondly, urban GS also have a direct impact on citizens’ resilience during pandemic outbreaks [12], particularly on marginalized groups like low-income populations [13]. Whether in a pandemic period or not, GS are linked to numerous public health benefits, both in term of mental and physical health [14]. These health benefits are partially explained by the psychological relaxation and stress reduction, the improvement of the psychological place attachment, the benefits to the immune system, and the enhancement of physical activity that GS promote [14]. During the COVID-19 pandemic, mental health has been a real public health issue, with symptoms of depression being much higher in the population than before the pandemic [15]. The importance of GS attendance was once again highlighted, studies showing higher probability of major depression by people who decreased their use of GS, compared to people whose use of GS increased or did not change [16]. Thirdly, and most importantly, GS contribute to the resilience of populations in times of crisis in an indirect way by fostering people’s social ties. A growing body of research suggests that natural environments play an important role in strengthening and enhancing our interpersonal relationships [9]. If a multitude of research has found that meaningful and high-quality connections with others are one of the most reliable predictors of happiness, wellbeing, and health [17,18,19], people’s social capital (i.e., the norms and ties among and between residents in communities [20]) seems even more important in times of crisis. People’s social capital represents an important resilient factor to overcome periods of crisis [21,22,23], given that connections with others provide information, resources, moral support [24,25], and create more positive recovery processes [26]. This third point is of particular interest to us in this research.

### 1.2. The Impact of Green Space Attendance on Prosocial Behaviors

One possible way to create, maintain, and strengthen connections with others, and, thus, improve our social capital, would be to assist others by adopting prosocial behaviors [27]. Prosociality refers to the tendency to care for, help, and assist others [9]. As already highlighted, people’s actions and behaviors, like their interactions with others, are not only influenced by their social environment (i.e., other people) but are also affected by a relatively ‘asocial’ natural environment [9]. Some experimental studies demonstrate that nature exposure can directly increase prosociality [28,29,30]. In a field experiment, passers-by who just walked across a park were more likely to help confederates who accidentally dropped a glove on the ground, than passers-by who were tested before entering the park [28]. Another study found out that, compared to sitting in a windowless laboratory room, sitting in a park boosted feelings of interconnectedness [29]. Even incidental exposure to nature in the lab, by looking at pictures of nature instead of pictures of urban environments can enhance prosociality [30]. Research documents two characteristics/qualities of natural environments that drive our orientation to others and their needs: feelings of awe [31] and perception of beauty [32,33]. Awe involves positively valenced feelings of wonder and amazement and, at least in Western cultures, comes up in encounters with nature like sunsets, scenic vistas, and mountain ranges [9,34]. However, most people do not have access to “awe-inspiring” GS on a daily basis, given that it seems almost impossible to find this type of landscape in urban environments. Fortunately, awe is not the only dimension that triggers increased social connection. The perception of beauty in natural environments can also increase prosocial tendencies [32,33]. Participants exposed to a beautiful nature report increased positivity and, as a consequence, behave more prosocial and are more willing to incur costs for the benefit of other participants [32,33]. In one set of experiments, participants who viewed beautiful nature pictures were more generous in an economic game than those who viewed more mundane nature images, and participants exposed to beautiful plants provided more help by constructing origami figures for tsunami victims than those exposed to more ordinary plants [33]. While we can assume that urban GS can hardly provide a feeling of awe, they can be perceived as beautiful and aesthetic and, therefore, appear to have the potential to contribute to the prosocial behaviors of individuals who attend them.

### 1.3. Crowdedness Perception of the GS

If a lot of research suggests that social interactions are positively influenced by GS’ presence and quality [35,36,37,38], such as the aesthetic and well-maintained appearance of the space mentioned before, it is important to highlight that these studies focus mostly on the physical aspects of the spaces. However, these spaces also include a whole “social” aspect due, for example, to the presence or absence of people, the interactions sought or avoided, or the activities that do or do not take place there. This social aspect seemed central to us given the pandemic situation and the resulting restrictions of social contacts. We often assume that urban GS are mainly seen as meeting places, given that they provide opportunities for people to interact with others in ways that may not occur in other settings [35]. Although these spaces are indeed important meeting places in the urban environment, it is important to highlight that, even if people tend to engage in small talk with other visitors, they generally do not visit parks with the intention to meet strangers [39]. Not seeking contact with strangers when visiting GS appeared even truer during the pandemic—with a significant reduction of activities that could be considered as high-risk activities such as meeting people [40]. A study conducted during the first COVID-19 lockdown in Croatia, Israel, Italy, Lithuania, Slovenia, and Spain, supports the importance to expand the role of GS beyond the fact of just creating and maintaining social bonds and emphasizes that parks and other urban GS have essential functions that are fundamentally different from other types of public places [40]. While urban GS can of course serve as a center of public gathering, they also meet vital needs of isolation from ambient urban stress and provide space to disconnect and relax [12,40]. In this period of a pandemic, it, therefore, seems possible that urban GS have mainly fulfilled a function of withdrawal from stressful environments (urban environment, family environment, overcrowded housing…) and responded primarily to an objective of restoration, rather than to an objective of socialization. This is also suggested by a study that compared the reasons for using GS before (2019) and during COVID-19 pandemic (2020), highlighting an increase in people’s self-reported need to use GS for ‘stress relief’ [16]. As mentioned before, natural environments can influence people’s prosocial behaviors [9], but the social environment seems equally important, particularly in the very specific context of the COVID-19 pandemic and the unprecedented measures of social restrictions associated with it. In this context, it is conceivable that uncrowded places allow stress reduction on individuals, which allows mood improvement and, thus, positively impacts the attitude toward others. On the opposite, overcrowded places can induce additional stress, not only because they do not allow the desired isolation, but also because of the increased risk of contamination that this represents. This reasoning is consistent with data from a study showing that the main self-reported reason for the non-use of GS among people who decreased GS attendance during lockdown is the fear and anxiety about coronavirus [16].

### 1.4. Hypotheses

The current study was designed to explore how GS attendance, perceived beauty of the GS, and perceived crowdedness of the GS, relate to social orientation during this specific time of health crisis. Specifically, we make the hypothesis that the positive relationship between GS attendance and prosocial behavior will only appear when the most regularly used GS is perceived as beautiful and uncrowded.

We also included several covariates. Perception of beauty and perception of the crowdedness of the visited GS requires that the individual pays attention to the environment that surrounds her or him. Using technologies like mobile phones when walking around outside may distract people from the beauty or the crowdedness of the space they are walking through, preventing total immersion in this natural space and, thus, their ability to savor their surroundings. Therefore, it seems important to consider the usage habits of this type of technology when going for a walk. Moreover, the lockdown conditions were by far not the same for everyone. Some families were confined to small apartments without balconies or gardens, while some others had homes that allowed them to find a certain balance between contact with other family members and moments of isolation. Some people experienced confinement alone, totally isolated from social interactions for a more or less long period. It seems obvious to take into account not only the number of people an individual was confined with, but also the perception of lockdown constraints, which can be experienced very differently from one individual to another, and which will more than likely affect his or her attitude towards others. Distance between home and the most attended GS was also controlled for, given that it has a direct impact on the regularity of use of the GS, particularly during the lockdown [40]. Finally, we controlled participants’ gender, since research documents a significant difference between men and women in prosocial behavior, probably due to a difference in the level of empathy, which tends to be higher for women [41].

## 2. Materials and Methods

A cross-sectional study using convenience and snowball sampling was conducted among French-speaking people in April 2020.

### 2.1. Participants and Procedure

Data were collected in April 2020, during the first COVID-19 lockdown, using an online survey. As a reminder, in April 2020, people were asked to remain confined to their homes and only essential daily commuting was permitted. Going out for a walk was allowed, provided that you did it on your own, accompanied only by people from your “social bubble” or a maximum of one person from outside this “social bubble”. Participation was voluntary and unpaid. Using G*Power [42], we calculated the number of participants needed for a power of 0.90 and a small effect size (*f*^2^ = 0.02, *α* = 0.05). Based on this, a minimum of 636 subjects was required. A total of 1206 participants (972 female, aged between 17 and 77 years, *M*_age_ = 28.74, *SD*_age_ = 12.87) participated in this study. Data from 610 participants (465 female, aged between 17 and 77 years, *M*_age_ = 28.63, *SD*_age_ = 12.75) were analyzed after removing those who indicated that they had not left their homes at all from the beginning of the lockdown or only for utility purposes (shopping, pharmacy, post office, etc.). Only participants who reported visiting a GS at least once (mainly urban parks, but also walking paths, shared gardens, etc.) were retained for further analysis. An invitation message with an online link to the survey was posted on social media (i.e., Facebook), explaining the objectives of the study and asking people to complete the survey and spread the message on their social platforms. Before starting the survey, confidentiality assurance was provided to the participants, who could only enter the survey after giving their online consent to participate. This study was approved by the Ethics Committee of the Faculty of Psychology, Speech and Language Therapy, and Education of the University of Liège (Ref. No. 1920-88, date of approval: 7 April 2020).

The survey started with a measure of prosocial behavior, in which participants were asked to allocate points to themselves and a hypothetical other. The real purpose of this measure was hidden from participants. To have a more ecological measure, participants were told that this first step was in no way related to the main objective of the study but was intended to validate a measurement tool that would be used in future studies. Participants were then informed that they were now moving on to the main study. In addition to the usual socio-demographic questions asking participants to indicate their gender, age, last obtained degree, and profession, participants were also asked to indicate the number of people confined with, the start day of their lockdown (knowing that the beginning of the lockdown was not the same in Belgium and in France), perceived lockdown constraints, and two questions regarding their confinement location (type of dwelling and whether they considered it to be in a rural or urban area). After these general questions, participants were asked to indicate how many times they attended a GS since the beginning of their lockdown. If participants indicated that they had visited a GS at least once, they were asked to indicate the number of times they had visited the GS they felt they had visited most since the beginning of their lockdown. Participants were then asked to evaluate the beauty and the perceived crowdedness of this specific GS, and to indicate if they were used to do several things at the same time when visiting this specific GS (e.g., playing on their phone). The study ended with a debriefing explaining the objectives of the study.

### 2.2. Materials

*Prosocial behavior.* Prosocial behavior was assessed using Social Value Orientation (SVO), given that SVO has been validated to be predictive of real-life prosocial behavior [43,44,45,46,47,48,49]. Therefore, we used the Social Value Orientation Slider Measure (SVO slider measure) [50]. This measurement tool produces a continuum of SVO instead of discrete categories, which destroys valuable information about individual differences [50,51]. This continuum reflects the degree to which a decision-maker will choose to sacrifice his or her resources to benefit another [51]. The slider measure is a widely used, efficient, and simple measurement of SVO. While this measure is typically conceptualized as a measure of individual difference, instructions of the SVO slider measure do not imply anything trait-like [52], and similar measures have been shown to be context-sensitive [53]. The SVO slider measure [49] asks participants to allocate points to themselves and a hypothetical other. The measure consists of six items, each representing a forced choice of nine alternatives that vary benefits to oneself vs. others. Collected responses allow calculating an “SVO angle” for each participant: “altruists” have very high SVO angles, “prosocials” have moderately high SVO angles, “individualists” have low SVO angles, and “competitors” have very low SVO angles. This means that the higher the score, the more prosocial the choices of participants are.

*Green space attendance*. Green spaces attendance was evaluated using the following instruction: “How many times do you estimate that you have used a green space since the start of lockdown? Example: urban park, forest, walking path with natural features… (If you can’t remember the exact number of times, give an approximate number. You can also write down an order of magnitude such as “every day”, “once a week”, etc.)”. Participants who had visited a GS at least once were then asked the same question about the most attended GS using the following instruction: “Think about the green space you visited most regularly. Since the beginning of the lockdown, how many times have you been to this green space? (If you can’t remember the exact number of times, give an approximate number. You can also write down an order of magnitude such as “every day”, “once a week, etc.). Since the data collection software provided the exact day of the participants’ response and since they were asked to indicate the start date of their confinement, it was possible to calculate the number of GS attendances per participant, for their entire lockdown. To make the data comparable between participants, the following formula was used: (GS attendance/lockdown length) * 100 with ‘GS attendance’ representing the number of times the participant attended a GS since the beginning of their confinement and ‘lockdown length’ representing the number of day participants were confined. This variable is named ‘GS (all) ‘for further analysis. The same ratio was applied to the most frequented GS: (GS attendance/lockdown length) * 100 with ‘GS attendance’ representing the number of times the participants attended their most visited GS since the beginning of their confinement and ‘lockdown length’ representing the number of day participants were confined. This variable is named ‘GS (main)’ for further analysis.

*Perceived beauty.* Participants were asked to rate the perceived beauty of the GS they had visited most regularly since the beginning of the lockdown. Beauty perception was assessed by 3 items, using bipolar scales ranging from −3 (unpleasant; ugly; inhospitable) to +3 (pleasant; beautiful; welcoming). The median value represents a neutral opinion. Internal consistency was sufficient (Cronbach *α* = 0.73).

*Perceived crowdedness.* Participants were asked to rate the perceived crowdedness of the GS they had visited most regularly since the beginning of the lockdown. Crowdedness perception was assessed by 3 items, using bipolar scales ranging from −3 (calm; quiet; lightly frequented) to +3 (lively; noisy; heavily frequented). The median value represents a neutral opinion. Internal consistency was sufficient (Cronbach *α* = 0.77).

*Covariates.* Distraction, number of people confined with, perceived lockdown constraint, the distance between home and the most attended GS, and gender were controlled statistically in our analyses. Distraction was assessed by asking participants to indicate whether they consulted their phones while visiting GS during this lockdown period. The frequency with which they consulted their phone while walking outside was measured on a 7-point Likert scale, with 1 indicating that they never used their phone and 7 indicating that they always used their phone while visiting urban GS. The number of people confined was measured by asking one single question: “Without counting you, how many people are confined with you?” Perceived lockdown constraint was assessed by asking participants to indicate, on a 7-point Likert scale, how restrictive they perceived the lockdown measures, with 1 indicating that they perceived the measures as being not at all restrictive and 7 indicating that they perceived the measures as being extremely restrictive. Distance between the most attended UGS and place of residence was assessed by one single question: “Approximately how long does it take you to walk to this location? (Note the time in minutes)”. Looking at the actual distance in meters would have been less relevant in the context of the current study, given that the time to walk the same distance depends on age and physical condition, among other things, and, thus, greatly influences accessibility to the space (the actual data we are interested in).

### 2.3. Statistical Analysis

Statistical analysis was performed using Jamovi (version 2.2.5) [54]. After analyses allowing to describe the sample, Spearman’s correlations were applied to examine the bivariate correlations between GS attendance, GS beauty perception, GS crowdedness perception, and prosocial behavior, given that none of the variables of interest was normally distributed. The associations of GS attendance, GS perception (beauty and crowdedness), and prosocial behavior were further analyzed using regression models (multiple linear regressions), which controlled potential confounders. Specifically, SVO slider measure score (prosocial behavior) was entered as dependent variable; beauty perception, crowdedness perception, the interactions between beauty perception and GS attendance, and the interaction between crowdedness perception and GS attendance were entered as independent variables; distraction, number of people confined with, perceived lockdown constraint, the distance between home and the most attended GS and also gender were entered as covariates. We ran a second separate regression model including both the covariates and the two-way interactions between the covariates and the two moderators (i.e., perception of beauty and perception of crowdedness) to avoid bias in the estimation of interaction effects [55]. For both regression models, independent variables were mean-centered, and the significance level was always set at *p*-value < 0.05.

## 3. Results

Our sample consists of two sub-samples, participants who have visited at least one GS since the beginning of their confinement and those who have not (Table 1). There is no significant difference in social orientation scores between participants who report having visited a GS at least once since the beginning of their lockdown and those who have not (*U_mann-whitney_* = 175,028, *p* = 0.26). A more in-depth analysis of these samples reveals a general significant difference in the use of GS according to the type of lockdown dwelling (*χ*^2^ = 12.37, *p* = 0.002, *df* = 2). There is a significant difference between the group of people who have access to a private garden and the group of people who have no private outdoor space (*W* = −4.15, *p* = 0.009), with people owning a garden attending GS more (*M_GS attendance_* = 21.99, *SD_GS attendance_* = 32.03) than people without outdoor space (*M_GS attendance_* = 13.33, *SD_GS attendance_* = 25.99). In addition, a general significant difference in the use of GS can be observed according to the level of education (*χ*^2^ = 13.24, *p* = 0.004, *dl* = 3). There is a significant difference between participants whose last obtained degree was the “CESS” (= “Certificat d’enseignement secondaire supérieur”—certificate of higher secondary education) preparing for advanced studies and participants whose last obtained degree was a restrictive professionalizing “CESS” giving access to technical or manual professions (*W* = −4.22, *p* = 0.015). Participants owning the professionalizing “CESS” attend GS less *(M_GS attendance_* = 14.20, *SD_GS attendance_* = 27.79) than participants owning the “CESS” preparing for advanced studies *(M_GS attendance_* = 21.16, *SD_GS attendance_* = 34.96). There is also a significant difference between participants whose last obtained degree was the professionalizing “CESS” and participants who went to college or university (*W* = 4.76, *p* = 0.004). Participants whose last obtained degree was the professionalizing “CESS” attend GS less *(M_GS attendance_* = 14.20, *SD_GS attendance_* = 27.79) than participants who went to college or university *(M_GS attendance_* = 23.09, *SD_GS attendance_* = 32.77). Finally, when taking the overall sample, a general significant difference in GS attendance can be observed between people who were confined alone and people confined with at least one other person (*U_mann-whitney_* = 32,720.5, *p* = 0.022). Participants confined alone attended GS less (*M_GS attendance_* = 17.17, *SD_GS attendance_* = 34.09) than participants confined with at least one other person (*M_GS attendance_* = 21.00, *SD_GS attendance_* = 33.08).

Table 2 shows Spearman’s rho and the significance of all tested bivariate correlations. None of the variables was significantly correlated to prosocial behavior. GS attendance was significantly correlated to the perceived beauty and to the perceived crowdedness of the place. Perceived beauty of the GS and perceived crowdedness of the GS were also significantly correlated.

Multiple linear regressions (Table 3) show that the interaction between GS attendance and perceived crowdedness of the place was significantly associated with prosocial behavior, after controlling for distraction, number of people confined with, perceived lockdown constraints, distance between home and the most attended GS, and gender. Number of people confined with and gender are also significantly associated with prosocial behavior. An increase in the number of people confined with is associated with fewer prosocial behaviors and female participants (*M*_prosocial_ = 30.47, *SD*_prosocial_ = 15.36) show more prosocial behaviors than male participants (*M*_prosocial_ = 25.49, *SD*_prosocial_ = 18.56).

Table 4 shows that the association between GS attendance and perceived crowdedness of the place is only significant at low crowdedness perception.

We ran the second separate regression model including both the main effect of covariates and two-way interactions between the covariates and the two moderators (i.e., perception of beauty and perception of crowdedness). This second regression model is outlined in the Appendix A (Table A1 and Table A2). If the interaction between the crowdedness perception and attendance rate of the GS is no longer statistically significant, it should be noted that the pattern is maintained (*b* = −0.029, *SE* = 0.016, *t* = −1.74, *p* = 0.082).

## 4. Discussion

The current study was designed to explore how GS attendance, perceived beauty of the GS and perceived crowdedness of the GS relate to social orientation in this specific time of health crisis. We assumed that the relationship between GS attendance and prosocial behaviors would be stronger when the most attended GS was perceived by the respondent as beautiful and uncrowded. Regression analyses revealed that the interaction between GS attendance and the perceived crowdedness of the place was significantly associated with prosocial behavior. After decomposition of the interaction, our results suggest a significant relationship between GS attendance and prosocial behavior but only when the crowdedness of the most visited GS was perceived as low. Contrary to the hypothesis, results did not show a significant relationship between beauty perception, attendance rate, and prosocial behavior. Finally, according to our regression analyses, female participants significantly scored higher on the prosocial measurement.

It seems important to take into account the variable of crowdedness perception as the interaction between low crowdedness perception and GS attendance is significantly associated with prosocial behavior. This result seems consistent with the latest theoretical advances on the subject and the results of previous studies. A growing number of research appears to support the fact that our use and perception of urban GS are influenced by external events, like cultural background or environmental factors, such as most probably the COVID-19 pandemic. The debate about whether human perception of nature is innate (evolutionary theories) or learned (cultural theories) is certainly far from being closed and future theories would benefit from combining evolutionary and cultural approaches [56]. Nevertheless, more and more studies underline the central role of external events on our perception of nature. This can be highlighted by studies showing that differences in perception and use exist between localities [57,58,59,60], or, for instance, by studies showing the influence of events such as the COVID-19 on people’s GS perception and use [40]. The use people make of GS changed during the pandemic, with a reduction of activities that possibly increase infection risk, like meeting strangers [40]. It underlines the importance of seeing GS not only as meeting places but also as places that offer disconnection and relaxation [12,40]. In addition, it seems that positive moods positively impact prosocial behaviors [61,62] and even reinforce mutually one another [63]. Positive mood refers here to feeling relaxed, energetic, enthusiastic, content, calm, or cheerful [63]. Low crowded GS provide a better opportunity to disconnect than crowded, noisy parks and represent, in this very specific time of the pandemic, fewer infection risks. Therefore, given that the motivations for using GS changed during the pandemic and were probably primarily aimed at allowing a disconnection and a moment of relaxation in this very stressful period, low crowded GS possibly helped to maintain positive moods, which may positively influence prosociality.

This reasoning is also supported by Samuelsson and colleagues [12], who argue that the absence of physical confinement combined with positively contributing factors of natural environments possibly help to provide relaxation and stress reduction. A complementary reading of this result can be made according to the Self-Categorization Theory [64,65], which suggests that people have multiple social identities, which vary in salience depending on the context. According to this theory, the way we perceive ourselves will, therefore, shift from a more personal to a more collective/social identity (and vice versa) depending on the situation [65]. Our responses and our desire for crowds can indeed vary greatly from one context to another and crowdedness can sometimes be experienced as very pleasant and even actively sought [66]. Self-Categorization Theory explains that our response to a crowd will vary according to our psychological proximity to the people in that crowd, i.e., whether or not those people are part of our in-group [66]. The more the members of this crowd are perceived as “other”, the more people will seek a spatial distance or, if this is impossible, will experience the situation as stressful or unpleasant [66]. It would not be surprising that the lockdown and its associated physical distancing, as well as the risk of infection represented by other individuals, accentuated the psychological distance we have with other people, thus making promiscuity undesirable and even extremely stressful. In this context, the presence of crowds generates negative emotions and stress, whereas sparsely populated spaces are associated with positive mood, which, as we saw earlier, impacts positively prosocial behaviors [61,62].

The presumed link between the perceived beauty of the place and prosocial behavior was not supported. Results did not show a significant relationship between beauty perception, attendance rate, and prosocial behavior. This is not in line with previous studies, which showed that beauty perception can trigger increased social connection [32,33]. However, the impact of beauty perception on prosocial behavior seems to be mediated by a positive mood [32,33]. A first possible explanation for this difference in results lies in the setting of the studies. In the studies quoted, participants were confronted with pictures of nature and the prosocial behavior was measured directly afterward. In the case of the present study, a cross-sectional design rather than an experimental study, there is no manipulation of GS exposure. Participants started with the social orientation measure and only afterward rate the GS they visited most since the beginning of the lockdown, so they were not confronted or primed with it right before answering the survey. No significant relation was found, perhaps because the effect of the beauty of the space does not hold over time. Another possible explanation is based on the results of previous studies highlighting that the impact of beauty on prosocial behavior seems mediated by positive feelings (happiness, joy, satisfaction, pleasure, fun…) [43]. In times of health crisis, it is possible that the beauty of the space is not sufficient to significantly impact this type of feeling, the average mood of the individuals being most probably lower than usual (floor effect).

Our results also highlighted a significant relationship between the number of people with whom participants were confined and prosocial behavior. The higher the number of people with whom participants were confined, the lower the SVO scores. It is conceivable, that being confined almost 24 h a day with other people depletes the “social energy” of an individual, who will then tend to show less prosocial behaviors. If during the lockdown, GS have mainly been visited to find places of isolation, allowing to “recharge one’s social battery”, then we should observe a less important attendance rate of GS among participants living alone, considering that they would not need to recharge this “social battery”. This reasoning is consistent with our results, as our analyses show less GS attendance among participants living alone. This, again, emphasizes the importance of not only seeing these spaces as meeting places but also as spaces of isolation.

In addition, our results highlighted a significant difference in prosocial behaviors between men and women, with women scoring higher on the SVO than men. This result is consistent with recent findings in this field, which seem to support that a difference in prosociality between genders would be due to the difference in empathy, higher in women [41]. However, it is important to note that Kamas and Preston’s study [41] is specifically based on economic games, which is congruent with the type of measure we used in the present study (SVO slider measure [51]). Seeing this behavior as complex and multidimensional allows taking into account different types of prosocial behaviors [67], as well as how prosocial behavior varies as a function of the target of the behavior (e.g., strangers vs. family members…) [67,68]. When prosocial behavior is seen as unidimensional, items tend to represent behaviors that are in line with the female stereotype of care [67]. Items representing behaviors that are more in line with masculine stereotypes, like physical assistance and helping behavior in case of emergency, are often not included in prosocial behavior measurements even though they are still valid prosocial behaviors [67]. Although the SVO slider measure is neutral in its formulation and does not explicitly convey the idea of empathy, kindness, and caring for others, it probably comes even less close to a formulation that might be considered as typically “masculine”, as the behavior measured is neither risky, public, strength intensive, nor collectively oriented.

Finally, it seems important to go back to the descriptive statistics of the sample. Various observations were made regarding GS attendance. Counter-intuitively, people without access to a private GS used public GS significantly less than people having a private garden. However, this observation makes sense, since we know that it is often the low-income groups that live without access to private GS [6]. In the same way, lower education level is generally used as an operationalization for low socioeconomic status [69], and our observations show that people whose last obtained degree was a diploma preparing for college or university or people who currently attend college or university use GS significantly more than people whose last obtained degree was a restrictive professionalizing degree preparing for a manual or technical profession. Low economic status, thus, seems to be linked to lower GS attendance. The real question then is, why do people with low socioeconomic status use GS less during their lockdown. These observations are actually in line with previous studies. Low-income groups mostly concentrate in certain neighborhoods [5]. In many parts of the world, studies observe lower accessibility and quality of urban GS in low socio-economic neighborhoods, compared to urban GS in high socio-economic neighborhoods [70,71,72,73]. Not using urban GS during the lockdown, even though one does not have access to a private garden, can, therefore, potentially be explained by the fact that these people did not have access to such spaces or that these spaces were of poor quality.

### 4.1. Practical Implications

Our results show a very small effect size for the interaction between crowdedness perception and GS attendance on prosocial behavior. However, small effects may have direct real-world consequences [74,75] and this is particularly true for effects that accumulate over time and at scale [74,76]. Thus, while small effects may not matter much for a single episode or a single individual, small effects matter in the long term and on a large scale [74,75,76]. Presently, more than half of the world’s population lives in urban areas and this number is continually increasing, leading to the estimation that in 2050 this will be the case for more than two-thirds of the world’s population [77]. Therefore, increasing the presence of urban GS will benefit a large population (large-scale consequence). Furthermore, given the many positive impacts of GS, prevention policies should aim to increase the regular use of GS by individuals, not just a “once in a while” use. The impact, even minimal, is, therefore, an impact that a single individual can accumulate throughout his or her whole life (long-term consequence). Given the large-scale consequences and the long-term consequences GS can have, this effect might, therefore, be highly consequential from a public-health perspective. It is also important to remember that, like many psychosocial phenomena, prosocial behaviors are complex and obviously multifactorial, which makes them unlikely to be explained by a few strong predictors with large effect sizes [78].

As described previously, people’s social relationships represent an important resilient factor to overcome periods of crisis [21,22,23]. Based on the results of this study, the opportunities for isolation and disconnection offered by GS seem to be related to the prosocial behaviors of their users, and, thus, probably contribute to the development of their social capital (i.e., the norms and ties among and between residents in communities [20]). If this may seem contradictory, offering oneself moments of isolation would possibly enable people to recharge their “social batteries”, allowing them to act in a more prosocial manner with others and so, on a long time, increase their social capital through quality interactions. Indeed, as mentioned before, assisting others through prosocial behaviors helps to improve our social capital [27]. The role GS have as places of isolation and disconnection seems particularly interesting to foster caring behaviors and a sense of community, which are crucial in times of crisis. Based on historical records, crisis situations seem to be a breeding ground for prosocial behaviors and feelings of a community [79,80,81], but providing individuals with an environment that allows the best conditions to encourage even more of this type of behavior, can only be a benefit to both the individual and the society. According to scientific predictions [2,3,4], we probably must expect an increase in pandemic outbreaks and, going with them, other periods of social distancing. In such a context, it seems important to invest in the multiplication of GS within urban landscapes, as well as thinking about their design. A well-thought-out design would probably reduce the feeling of overcrowding, even in very small GS.

As already highlighted, the results of this study, which are in line with many other studies pointing out the potentially positive impact of GS on populations, can have concrete implications for urban planning and management policies and so have a significant impact on public health. It seems currently accepted that COVID-19 is hitting the hardest lower-income groups [5]. During the pandemic, poverty and wage inequality raised in all European countries [82]. Bearing this in mind, cities need to find ways to function during these disturbances and to provide their most vulnerable populations with the necessary tools to cope as best as possible with such crises. Maintaining or increasing spaces for nature, while keeping it accessible to the public, seems to be part of it [12]. Creating urban landscapes that promote contact with nature, while allowing social distancing, can be achieved by a well-designed spatial organization [12,83], for instance by avoiding mono-functional high-density areas and increasing accessible natural spaces [84]. However, it is also important to keep in mind that property-rights arrangements are equally important to consider [85]. Currently, public accessible GS are not equally available to all population groups [7,8], and the average distance to access them increases with the poverty of a district [71]. This lack of access to quality publicly accessible GS among disadvantaged populations is all the more problematic since some studies seem to show that these populations would benefit the most from regular GS attendance [86,87,88]. Enabling low-income groups to access quality GS while maintaining social distancing will, thus, not only depend on whether such spaces actually exist in their neighborhood and how they are designed, but also on the fact that these spaces are private, public, or common ownership [12], and, therefore, will mainly depend on political decisions. Current privatization schemes can, therefore, lead to a gradual loss of opportunities or nature experiences for lower-income groups and, this way, result in undesirable societal outcomes [85]. On the contrary, investing in existing GS and creating new public GS in disadvantaged neighborhoods can have a variety of beneficial societal outcomes and seems to be an attractive strategy for health equity in pandemic recovery [13,89].

### 4.2. Study Limitations and Future Research

This study has several limitations, which open up possibilities for future research. First, given the cross-sectional design, the present study cannot make causal inferences. Ideally, an experimental study manipulating a green space’s vs. urbanized space’s crowdedness should be conducted in addition.

Second, as suggested in the discussion part, prosociality should be seen as multidimensional [67]. The SVO slider measure is one-dimensional and cannot account for the multitude of prosocial behaviors that exist. Therefore, it is possible that a measure that considers the multidimensionality of prosocial behaviors would produce more nuanced results.

Moreover, prosocial behaviors vary depending on the target person [68]. In the case of the present study, the results only concern prosocial behavior towards a stranger and, therefore, cannot be extended to a situation with friends or family members.

In addition, data do not allow us to know to what extent the perception is driven by pandemic or by pre-pandemic constructs. The need for uncrowded GS may be either a need related to the pandemic situation or a need that existed long before. Some studies comparing pre- and post-pandemic situations show that the reason for visiting GS changed during the pandemic, with a decrease in activities that could be considered as high-risk activities such as meeting people [40], and an increase in GS attendance for stress relief [16]. Considering that the main reason for the decrease of GS attendance seems to be the fear of the coronavirus (and not other reasons like the closure of the GS, governments’ incentives to stay home, or the lack of need to go out for example [16], it is possible that the perception, needs, and behaviors towards these spaces are different outside the pandemic period. It would, therefore, be interesting to replicate this study outside of a pandemic situation.

Last but not least, given that the sample is only composed of French-speaking people, it is possible that the obtained results are only applicable to Westernized cultures. As already mentioned, human perceptions and preferences towards natural environments significantly differ between countries [59,60], indicating that environmental factors like COVID-19, but also cultural background can condition the perception, expectations, and behaviors toward nature [40,56]. This cultural aspect, therefore, potentially influences our results.

## 5. Conclusions

In times of crisis, social ties can literally be a lifeline. A way to create, maintain, and strengthen connections and links between people is by assisting others by adopting prosocial behaviors. The current study was designed to explore how GS attendance, perceived beauty of the GS, and perceived crowdedness of the GS relate to social orientation (used as a proxy for prosocial behavior) in this very specific time of health crisis. Results showed a significant relationship between GS attendance and prosocial behavior, but only when the crowdedness of the most visited GS was perceived as low. Results did not show a significant relationship between beauty perception, attendance rate, and prosocial behavior. Finally, according to our results, female participants significantly scored higher on prosocial measurement. These results seem to support the fact that our use and perception of nature are influenced by external events, like the COVID-19 pandemic, and underline the importance of GS to fulfill vital needs of isolation and disconnection. Thereby, the present study contributes to a better understanding of the resilience role GS can play in times of crisis, shows the importance of increasing the availability of GS and allows concrete recommendations for public policies.

## Figures and Tables

**Table 1 ijerph-19-06778-t001:** Participants’ characteristics.

Participants Attending GS (N = 610)		Participants not Attending GS (N = 596)	
Characteristic	*M* ± *SD* or *n* (%)	Characteristic	*M* ± *SD* or *n* (%)
Age (year)	28.63 ± 12.75	Age (year)	28.85 ± 13.0
Gender (female)	465 (76.23)	Gender (female)	507 (85.07)
LD area (urban)	183 (30.0)	LD area (urban)	275 (46.14)
LD residence		LD residence	
Garden	514 (84.26)	Garden	464 (77.85)
Balcony	38 (6.23)	Balcony	54 (9.06)
No outdoor space	58 (9.51)	No outdoor space	78 (13.09)
Job		Job	
Student	389 (63.77)	Student	380 (63.76)
(Self-)employed	184 (30.16)	(Self-)employed	165 (27.68)
Unemployed	12 (1.97)	Unemployed	11 (1.85)
Unable to work	1 (0.16)	Unable to work	9 (1.51)
Retired	16 (2.62)	Retired	16 (2.68)
Other	8 (1.31)	Other	15 (2.52)
Degree		Degree	
No CESS	29 (4.8)	No CESS	40 (6.7)
CESS for HE	245 (40.2)	CESS for HE	215 (36.1)
CESS profess.	65 (10.7)	CESS profess.	94 (15.8)
HE	271 (44.4)	HE	247 (41.4)
PPL conf.	2.58 ± 1.42	PPL conf.	2.47 ± 1.69
LD constraint	4.22 ± 1.44	LD constraint	4.13 ± 1.54
SVO	29.29 ± 16.3	SVO	30.99 ± 14.90
GSA			
GSA (all-ratio)	41.09 ± 36.57		
GSA (main-ratio)	10.53 ± 9.57		
GSA (all-raw)	11.99 ± 10.76		
GSA (main-raw)	36.10 ± 32.34		
PB	6.3 ± 0.77		
PC	2.36 ± 1.35		
Distance	9.57 ± 13.91		
Distraction	2.73 ± 2.1		

LD area = area where participants were confined (urban; rural); LD residence = residence where participants were confined (with garden; with balcony without garden; no outdoor space); no CESS = no certificate of higher secondary education; CESS for HE = certificate of higher secondary education preparing to college or university; CESS profess.= certificate of higher secondary education preparing for technical or manual professions; HE = graduate or undergraduate; PPL conf. = number of people confined with; LD constraint = perceived lockdown constrain; SVO = prosocial behavior; GSA = green space attendance; GSA (all-ratio) = green space attendance using the ratio of all attended green spaces divided by the length of lockdown multiplied by hundred; GSA (main-ratio) = green space attendance using the ratio of the most attended green spaces divided by the length of lockdown multiplied by hundred; GSA (all-raw) = green space attendance of all attended green spaces; GSA (main-raw) = green space attendance of the most attended green spaces; PB = perceived beauty of the most attended green space; PC = perceived crowdedness of the most attended green space.

**Table 2 ijerph-19-06778-t002:** Correlation matrix between prosocial behaviors, GS attendance, perceived beauty of the GS, and perceived crowdedness of the GS.

	SVO	GSA	PB	PC
SVO	-			
GSA	*r_s_* = −0.001 *p* = 0.971	-		
PB	*r_s_* = 0.028 *p* = 0.488	*r_s_* = 0.145 *p* < 0.001 **	-	
PC	*r_s_* = −0.028 *p* = 0.483	*r_s_* = −0.086 *p* = 0.034 *	*r_s_* = −0.392 *p* < 0.001 **	-

SVO = prosocial behavior; GSA = GSA (main)—attendance rate of the most visited GS; PB = perceived beauty of the most visited GS; PC = perceived crowdedness of the most visited GS. * *p* < 0.05; ** *p* < 0.001.

**Table 3 ijerph-19-06778-t003:** Multiple linear regressions assessing associations between GS attendance, GS perception, the interaction between both, and prosocial behaviors (dependent variable), while controlling for distraction, number of people confined with, perceived lockdown constraint, distance, and gender.

			95% IC					
	*b*	*SE*	Lower	Upper	β	*df*	*t*	*p*
(Intercept)	27.75	0.770	26.24	29.26	0.000	599	36.06	<0.001
GSA	0.018	0.021	−0.023	0.060	0.036	599	0.860	0.390
PB	−0.817	0.924	2.632	0.999	−0.039	599	−0.884	0.377
PC	−0.413	0.520	−1.435	0.609	−0.034	599	0.343	0.732
GSA * PB	−0.003	0.030	−0.063	0.057	−0.005	599	−0.104	0.917
GSA * PC	−0.034	0.017	−0.066	−0.001	−0.090	599	−2.040	0.042 *
Distraction	−0480	0.310	−1.088	0.128	−0.062	599	−1.549	0.122
PPL conf.	−1.390	0.461	−2.294	−0.485	−0.121	599	−3.017	0.003 **
LD constraint	−0.576	0.453	−1.467	0.314	−0.051	599	−1.271	0.204
Distance	0.016	0.047	−0.076	0.108	0.014	599	0.343	0.732
Gender	−5.288	1.550	−8.332	−2.244	−0.324	599	−3.411	<0.001 **

GSA = GSA (main)—attendance rate of the most visited GS; PB = perceived beauty of the space; PC = perceived crowdedness of the space; GSA * PB = interaction between green space attendance and perceived beauty; GSA * PC = interaction between green space attendance and perceived crowdedness; PPL conf. = number of people confined with; LD constraint = perceived lockdown constraint. * *p* < 0.05; ** *p* < 0.001.

**Table 4 ijerph-19-06778-t004:** Simple effects of GS attendance on prosocial behaviors at high, middle, and low perceived crowdedness of the place.

Moderator Levels			95% IC					
PC	*b*	*SE*	Lower	Upper	β	*dl*	*t*	*p*
Mean − 1.*SD*	0.064	0.029	0.006	0.121	0.126	599	2.174	0.030 *
Mean	0.018	0.021	−0.023	0.060	0.036	599	0.860	0.390
Mean + 1.*SD*	−0.027	0.032	−0.090	0.036	−0.054	599	−0.850	0.396

PC = perceived crowdedness of the space. * *p* < 0.05.

## Data Availability

The data presented in this study are openly available in OSF, reference number osf.io/8a634.

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
