# Peer review of "Relationships between Green Space Attendance, Perceived Crowdedness, Perceived Beauty and Prosocial Behavior in Time of Health Crisis"

_ijerph, 2022, doi:10.3390/ijerph19116778_

Round 1
Reviewer 1 Report
My main suggestion is to shorten the introduction that is a bit too long and to make a deeper analysis of the most recent literature. Besides, in the introduction section, the author mentioned covid-19, however, in the result part, it is not related to it for analysis. Meanwhile, the statistical analysis methods should be described in detail, and each part of the methodology is expected to be described separately and comprehensively. Besides, some knowledge and methodological backgrounds were not presented in the introduction and methodology but with results.
Author Response
Thank you for your feedback and the useful improvement suggestions. Please find here below our responses, point-by-point. Attached you will find the final manuscript, with the modifications included (tracking tool).
Point 1: My main suggestion is to shorten the introduction that is a bit too long and to make a deeper analysis of the most recent literature.
Response 1: Thank you for your feedback and the useful improvement suggestions. Due to the remarks of other reviewers, we were unable to reduce the "introduction" part. However, we have added two recent references :
Page 2, line 80 ; Page 4, line 149 and line 158 (reference number 16) :
Heo, S.; Desai, M.U.; Lowe, S.R.; Bell, M.L. Impact of Changed Use of Greenspace during COVID-19 Pandemic on Depression and Anxiety. Int. J. Environ. Res. Public Health 2021, 18, 5842.
Page 2, line 76 (reference number 15) :
Ettman, C.K.; Abdalla, S.M.; Cohen, G.H.; Sampson, L.; Vivier, P.M.; Galea, S. Prevalence of Depression Symptoms in US Adults Before and During the COVID-19 Pandemic. JAMA Netw. Open 2020, 3, e2019686
Point 2: Besides, in the introduction section, the author mentioned covid-19, however, in the result part, it is not related to it for analysis.
Response 2: We accounted for the pandemic situation in the analyses by including the covariates of "perceived lockdown constraints" and "number of people participants were confined with" (see also Table 3). Concerning the descriptive analyses of the samples, we also looked at the differences in the use of green spaces according to the conditions of confinement (dwelling type, see Table 1; LD residence = residence where participants were confined (with garden; with balcony without garden; no outdoor space). We agree, however, that it is not possible to know from these results whether our results are due to the pandemic situation or not. We decided to include this point in the limits of the study (see page 14, line 590 to line 618). Here below, the addition we have made in the limitation part (added elements are in bold) :
“ This study has several limitations, which open up possibilities for future research. First, given the cross-sectional design, the present study cannot make causal inferences. Ideally, an experimental study manipulating a green space’s vs. urbanized space’s crowdedness should be conducted in addition.
Second, as suggested in the discussion part, prosociality should be seen as multidimensional [65]. The SVO slider measure is one-dimensional and cannot account for the multitude of prosocial behaviors that exist. Therefore, it is possible that a measure that considers the multidimensionality of prosocial behaviors would produce more nuanced results.
Moreover, prosocial behaviors vary depending on the target person [66]. In the case of the present study, the results only concern prosocial behavior towards a stranger and therefore cannot be extended to a situation with friends or family members.
In addition, data do not allow us to know to what extent the perception is driven by pandemic or by pre-pandemic constructs. The need for uncrowded GS may be either a need related to the pandemic situation or a need that existed long before. Some studies comparing pre- and post-pandemic situations show that the reason for visiting GS changed during the pandemic, with a decrease in activities that could be considered as high-risk activities such as meeting people [40], and an increase in GS attendance for stress relief [16]. Considering that the main reason for the decrease of GS attendance seems to be the fear of the coronavirus (and not other reasons like the closure of the GS, governments’ incentives to stay home, or the lack of need to go out for example [16], it is possible that the perception, needs, and behaviors towards these spaces are different outside the pandemic period. It would therefore be interesting to replicate this study outside of a pandemic situation.
Last but not least, given that the sample is only composed of French-speaking people, it is possible that the obtained results are only applicable to Westernised cultures. As already mentioned, human perceptions and preferences towards natural environments significantly differ between countries [57,58], indicating that environmental factors like COVID-19, but also cultural background can condition the perception, expectations, and behaviors toward nature [40,54]. This cultural aspect therefore potentially influences our results.”
Point 3: Meanwhile, the statistical analysis methods should be described in detail, and each part of the methodology is expected to be described separately and comprehensively.
Point 4: Besides, some knowledge and methodological backgrounds were not presented in the introduction and methodology but with results.
Responses 3 and 4: Thank you for this feedback, we have indeed put elements of the “Statistical Analysis” in the results and we also agree that this “Statistical Analysis” is consequently incomplete. We have made the changes accordingly. Below you can read the rewritten "Statistical Analysis" part. The removal of the elements of the “Statistical Analysis” part in the “Results” part is visible directly in the manuscript (thanks to the tracking tool) in the attachment. Here below, the addition we have made in the limitation part.
Starting page 6 and ending page 7 (line 301 to line 318), items in bold represent information removed from the results section and newly added to this “Statistical Analysis” section:
“Statistical analysis was performed using Jamovi (version 2.2.5) [54]. After describing the sample, Spearman’s correlations were applied to examine the bivariate correlations between GS attendance, GS beauty perception, GS crowdedness perception, and prosocial behavior, given that none of the variables of interest was normally distributed. The associations of GS attendance, GS perception (beauty and crowdedness), and prosocial behavior were further analyzed using regression models (multiple linear regressions), which controlled potential confounders. Specifically, SVO slider measure score (prosocial behavior) was entered as dependent variable; beauty perception, crowdedness perception, the interactions between beauty perception and GS attendance, and the interaction between crowdedness perception and GS attendance were entered as independent variables; distraction, number of people confined with, perceived lockdown constraint, the distance between home and the most attended GS and also gender were entered as covariates. We ran a second separate regression model including both the covariates and the two-way interactions between the covariates and the two moderators (i.e. perception of beauty and perception of crowdedness) to avoid bias in the estimation of interaction effects [48]. For both regression models, independant variables were mean-centered and the significance level was always set at p-value < .05.”

Reviewer 2 Report
The topic of the paper is interdisciplinary, recent, and has a broad range of practical implications.
The introduction chapter is well-elaborated with a clear structure. The major literature is cited from the realm of social ecology, pandemics, human behavior, and urban sociology studies. Although a broad spectrum of sciences is included, the theoretical background is coherent. As the research concentrates on urban areas, it would be advised to define the concept. It can be read even from this section, that the results can be applied only to Westernised cultures.
From the materials and methods part, this scope is even tightened only to French-speaking Europeans. The sampling method seems to be proper, but it would be great to describe the way of the survey in detail.
The materials section is well elaborated, so the basic concepts are explained properly. As the role of public green spaces was examined, a question may arise in the reader's mind about the possible role of private green spaces too.
The results are greatly summarized by the tables. As there are two "Table 2.", it is disturbing.
The first paragraph of the discussion should be enclosed in the conclusion chapter. At the same time, the discussion chapter is greatly framed with the theoretical background. The practical implication seems to be a bit broad as for example the role of social capital is mentioned without an explanation of the concept. The lack of access to quality green spaces among disadvantaged people should be mentioned in the earlier chapters too.
The conclusions part is quite short
Author Response
Thank you for your feedback and the useful improvement suggestions. Please find here below our responses, point-by-point. Attached you will find the final manuscript, with the modifications included (tracking tool).
Point 1: The topic of the paper is interdisciplinary, recent, and has a broad range of practical implications. The introduction chapter is well-elaborated with a clear structure. The major literature is cited from the realm of social ecology, pandemics, human behavior, and urban sociology studies. Although a broad spectrum of sciences is included, the theoretical background is coherent. As the research concentrates on urban areas, it would be advised to define the concept.
Response 1: Thank you for your feedback and all the useful suggestions and comments. We have, indeed, forgotten to define this concept! We added the definition on line 46. Below, the re-write of this part :
“Public urban GS consist mainly of semi-natural areas, referring in the present study to any vegetation found in the urban environment (i.e. towns, cities, suburbs, and their surroundings, which are characterized by high population density and built environment infrastructure), accessible to everyone without restriction (e.g. parks, playgrounds, walking paths, yards, plazas, peri-urban forests, road and rail networks, and their associated land, etc.).”
Point 2: It can be read even from this section, that the results can be applied only to Westernised cultures. From the materials and methods part, this scope is even tightened only to French-speaking Europeans.
Response 2: It is true that this is an aspect that we have not discussed and we agree that it is important to address it. We rewrote the “Study limitations and Future research” part in order to include this aspect (see page 14, line 615 to line 621). Below, the re-write of the “Study limitations and Future research” part (elements in bold are added elements) :
“This study has several limitations, which open up possibilities for future research. First, given the cross-sectional design, the present study cannot make causal inferences. Ideally, an experimental study manipulating a green space’s vs. urbanized space’s crowdedness should be conducted in addition.
Second, as suggested in the discussion part, prosociality should be seen as multidimensional [65]. The SVO slider measure is one-dimensional and cannot account for the multitude of prosocial behaviors that exist. Therefore, it is possible that a measure that considers the multidimensionality of prosocial behaviors would produce more nuanced results.
Moreover, prosocial behaviors vary depending on the target person [66]. In the case of the present study, the results only concern prosocial behavior towards a stranger and therefore cannot be extended to a situation with friends or family members.
In addition, data do not allow us to know to what extent the perception is driven by pandemic or by pre-pandemic constructs. The need for uncrowded GS may be either a need related to the pandemic situation or a need that existed long before. Some studies comparing pre- and post-pandemic situations show that the reason for visiting GS changed during the pandemic, with a decrease in activities that could be considered as high-risk activities such as meeting people [40], and an increase in GS attendance for stress relief [16]. Considering that the main reason for the decrease of GS attendance seems to be the fear of the coronavirus (and not other reasons like the closure of the GS, governments’ incentives to stay home, or the lack of need to go out for example [16], it is possible that the perception, needs, and behaviors towards these spaces are different outside the pandemic period. It would therefore be interesting to replicate this study outside of a pandemic situation.
Last but not least, given that the sample is only composed of French-speaking people, it is possible that the obtained results are only applicable to Westernised cultures. As already mentioned, human perceptions and preferences towards natural environments significantly differ between countries [57,58], indicating that environmental factors like COVID-19, but also cultural background can condition the perception, expectations, and behaviors toward nature [40,54]. This cultural aspect therefore potentially influences our results.”
Point 3: The sampling method seems to be proper, but it would be great to describe the way of the survey in detail.
Response 3: We have provided more details about the procedure (page 5, line 212 to line 231). Here below, the re-write of the procedure part (added elements in bold):
“(…)The survey started with a measure of prosocial behavior, in which participants were asked to allocate points to themselves and a hypothetical other. The real purpose of this measure was hidden from participants. To have a more ecological measure, participants were told that this first step was in no way related to the main objective of the study but was intended to validate a measurement tool that would be used in future studies. Participants were then informed that they were now moving on to the main study. In addition to the usual socio-demographic questions asking participants to indicate their gender, age, last obtained degree and profession, participants were also asked to indicate the number of people confined with, the start day of their lockdown (knowing that the beginning of the lockdown was not the same in Belgium and in France), perceived lockdown constraints and two questions regarding their confinement location (type of dwelling and whether they considered it to be in a rural or urban area). After these general questions, participants were asked to indicate how many times they attended a GS since the beginning of their lockdown. If participants indicated that they had visited a GS at least once, they were asked to indicate the number of times they had visited the GS they felt they had visited most since the beginning of their lockdown. Participants were then asked to evaluate the beauty and the perceived crowdedness of this specific GS, and to indicate if they were used to do several things at the same time when visiting this specific GS (e.g., playing on their phone). The study ended with a debriefing explaining the objectives of the study.
The survey started with a measure of prosocial behavior, in which participants were asked to allocate points to themselves and a hypothetical other. The real purpose of this measure was hidden from participants. To have a more ecological measure, participants were told that this first step was in no way related to the main objective of the study but was intended to validate a measurement tool that would be used in future studies. Participants were then informed that they were now moving on to the main study. In addition to the usual socio-demographic questions asking participants to indicate their gender, age, last obtained degree and profession, participants were also asked to indicate the number of people confined with, the start day of their lockdown (knowing that the beginning of the lockdown was not the same in Belgium and in France), perceived lockdown constraints and two questions regarding their confinement location (type of dwelling and whether they considered it to be in a rural or urban area). After these general questions, participants were asked to indicate how many times they attended a GS since the beginning of their lockdown. If participants indicated that they had visited a GS at least once, they were asked to indicate the number of times they had visited the GS they felt they had visited most since the beginning of their lockdown. Participants were then asked to evaluate the beauty and the perceived crowdedness of this specific GS, and to indicate if they were used to do several things at the same time when visiting this specific GS (e.g., playing on their phone). The study ended with a debriefing explaining the objectives of the study.”
Point 4: The materials section is well elaborated, so the basic concepts are explained properly. As the role of public green spaces was examined, a question may arise in the reader's mind about the possible role of private green spaces too.
Response 4: It is indeed a remark that makes sense, we decided to explain this choice in the “Introduction” part (see page 2, line 50 to line 53) :
“ (…) Public urban GS consist mainly of semi-natural areas, referring in the present study to any vegetation found in the urban environment (i.e., towns, cities, suburbs, and their surroundings, which are characterized by high population density and built environment infrastructure) accessible to everyone without restriction (e.g., parks, playgrounds, walking paths, yards, plazas, peri-urban forests, road and rail networks, and their associated land, etc.). Focusing on public accessible nature seems important, given that we know that it is often the poorest groups who live without access to private gardens [6]. However, it is important to keep in mind that currently, public accessible GS are not equally available to all population groups [7,8].”
Point 5: The results are greatly summarized by the tables. As there are two "Table 2.", it is disturbing.
Response 5: We have corrected this oversight and reviewed all table numbering, thank you for pointing it out!
Point 6: The first paragraph of the discussion should be enclosed in the conclusion chapter. At the same time, the discussion chapter is greatly framed with the theoretical background.
Response 6: Thanks a lot for the feedback on the discussion chapter! We have made the change as suggested and repeated this paragraph, slightly rephrased, in the conclusion chapter. The conclusion has also been fleshed out. Here below, the re-write of the conclusion part (Page 14, line 623 to line 637) :
“In times of crisis, social ties can literally be a lifeline. A way to create, maintain and strengthen connections and links between people is by assisting others by adopting prosocial behaviors. The current study was designed to explore how GS attendance, perceived beauty of the GS and perceived crowdedness of the GS relate to social orientation (used as a proxy for prosocial behavior) in this very specific time of health crisis. Results showed a significant relationship between GS attendance and prosocial behavior, but only when the crowdedness of the most visited GS was perceived as low. Results did not show a significant relationship between beauty perception, attendance rate, and prosocial behavior. Finally, according to our results, female participants significantly scored higher on prosocial measurement. These results seem to support the fact that our use and perception of nature are influenced by external events, like the COVID-19 pandemic, and underline the importance of GS to fulfill vital needs of isolation and disconnection. Thereby, the present study contributes to a better understanding of the resilience role GS can play in times of crisis, shows the importance of increasing the availability of GS and allows concrete recommendations for public policies.”
Point 7: The practical implication seems to be a bit broad as for example the role of social capital is mentioned without an explanation of the concept.
Response 7: We defined the concept of social capital in point 1.1. “Green spaces as resilience infrastructures” (see page 2, line 86), but to facilitate the reading and understanding of the “practical implications” part we repeated the definition at page 13, line 548 (see rephrasing here below).
“As described previously, people’s social relationships represent an important resilient factor to overcome periods of crisis [16–18]. Based on the results of this study, the opportunities for isolation and disconnection offered by GS seem to be related to the prosocial behaviors of their users, and thus probably contribute to the development of their social capital (i.e. the norms and ties among and between residents in communities [15])”.
Point 8: The lack of access to quality green spaces among disadvantaged people should be mentioned in the earlier chapters too.
Response 8: We have taken this suggestion into account, and have mentioned it in the introduction part (see page 1, line 41 to line 43). Here below, the re-write of this part of the introduction (added elements in bold) :
“We probably have to be prepared for other pandemics in the next coming years. In the current context, it is important to reflect on how to improve our capacity to deal with such crises, while considering that solutions have to be sustainable, i.e., that they consider both human and environmental aspects. Taking into account the social impact of the envisaged solutions means also paying particular attention to lower-income groups as it now seems accepted that COVID-19 is hitting these groups the hardest [5]. In this study, we would like to highlight the potential of publicly accessible urban green spaces (GS) to help face the consequences of crises such as pandemics. Public urban GS consist mainly of semi-natural areas, referring in the present study to any vegetation found in the urban environment (i.e., towns, cities, suburbs, and their surroundings, which are characterized by high population density and built environment infrastructure) accessible to everyone without restriction (e.g., parks, playgrounds, walking paths, yards, plazas, peri-urban forests, road and rail networks, and their associated land, etc.). Focusing on public accessible nature seems important, given that we know that it is often the poorest groups who live without access to private gardens [6]. However, it is important to keep in mind that currently, public accessible GS are not equally available to all population groups [7,8].
Point 9: The conclusions part is quite short
Response 9: The conclusion has been fleshed out (see point 6).

Reviewer 3 Report
ijerph-1737076-peer-review-v1
Relationships between Green Space Attendance, Perceived Crowdedness, Perceived Beauty and Prosocial Behavior in Time of Health Crisis
The authors have provided a great lit review. They may wish to also consider the following papers for framing and for the implications section:
Heo, S.; Desai, M.U.; Lowe, S.R.; Bell, M.L. Impact of Changed Use of Greenspace during COVID-19 Pandemic on Depression and Anxiety. Int. J. Environ. Res. Public Health 2021, 18, 5842.
Ettman, C.K.; Abdalla, S.M.; Cohen, G.H.; Sampson, L.; Vivier, P.M.; Galea, S. Prevalence of Depression Symptoms in US Adults Before and During the COVID-19 Pandemic. JAMA Netw. Open 2020, 3, e2019686
Spennemann, D.H.R. Exercising under COVID-2x: Conceptualizing Future Green Spaces in Australia’s Neighborhoods. Urban Sci. 2021, 5, 93. https://doi.org/ 10.3390/urbansci5040093
Meiring, R.M.; Gusso, S.; McCullough, E.; Bradnam, L. The effect of the COVID-19 pandemic movement restrictions on selfreported physical activity and health in New Zealand: A cross-sectional survey. Int. J. Environ. Res. Public Health 2021, 18, 1719.
Methodology:
Line 187 which Social media?
Line 191-193 Nice to see that Ethics approval is stated front and center !
Too late now, but it would have of interest to also query the respondents whether their perception of the green space had changed in time of COVID. As it stand, we have little idea to what extent the perceptions are driven by COVID or by pre-COVID constructs. This needs to be discussed
MINOR ISSUES
The manuscript needs to be read by a NATIVE English speaker as there are some awkward coices of words, presumably due to the fact that the authors are Francophone.
Author Response
Thank you for your feedback and the useful improvement suggestions. Please find here below our responses, point-by-point. Attached you will find the final manuscript, with the modifications included (tracking tool).
Point 1: The authors have provided a great lit review. They may wish to also consider the following papers for framing and for the implications section:
Heo, S.; Desai, M.U.; Lowe, S.R.; Bell, M.L. Impact of Changed Use of Greenspace during COVID-19 Pandemic on Depression and Anxiety. Int. J. Environ. Res. Public Health 2021, 18, 5842.
Ettman, C.K.; Abdalla, S.M.; Cohen, G.H.; Sampson, L.; Vivier, P.M.; Galea, S. Prevalence of Depression Symptoms in US Adults Before and During the COVID-19 Pandemic. JAMA Netw. Open 2020, 3, e2019686
Spennemann, D.H.R. Exercising under COVID-2x: Conceptualizing Future Green Spaces in Australia’s Neighborhoods. Urban Sci. 2021, 5, 93. https://doi.org/ 10.3390/urbansci5040093
Meiring, R.M.; Gusso, S.; McCullough, E.; Bradnam, L. The effect of the COVID-19 pandemic movement restrictions on selfreported physical activity and health in New Zealand: A cross-sectional survey. Int. J. Environ. Res. Public Health 2021, 18, 1719.
Response 1: Thank you very much for your feedback and these relevant paper suggestions! We added some of these references in our introduction part and in the practical implications.
Page 2, line 78; Page 3, line 148; Page 4, line 158 (reference number 16) :
Heo, S.; Desai, M.U.; Lowe, S.R.; Bell, M.L. Impact of Changed Use of Greenspace during COVID-19 Pandemic on Depression and Anxiety. Int. J. Environ. Res. Public Health 2021, 18, 5842.
Page 2, line 76 (reference number 15) :
Ettman, C.K.; Abdalla, S.M.; Cohen, G.H.; Sampson, L.; Vivier, P.M.; Galea, S. Prevalence of Depression Symptoms in US Adults Before and During the COVID-19 Pandemic. JAMA Netw. Open 2020, 3, e2019686
Page 13, line 573 (reference number 81) :
Spennemann, D.H.R. Exercising under COVID-2x: Conceptualizing Future Green Spaces in Australia’s Neighborhoods. Urban Sci. 2021, 5, 93. https://doi.org/ 10.3390/urbansci5040093
Point 2: Methodology: Line 187 which Social media?
Response 2: We have specified the media, i.e. Facebook, see page 5, line 205.
Point 3: Too late now, but it would have of interest to also query the respondents whether their perception of the green space had changed in time of COVID. As it stand, we have little idea to what extent the perceptions are driven by COVID or by pre-COVID constructs. This needs to be discussed
Response 3: This is indeed a relevant remark. We decided to include and discuss it in the "Study limitations" section (page 14, line 591 to line 621). Here below the re-write of this part (added elements in bold) :
“This study has several limitations, which open up possibilities for future research. First, given the cross-sectional design, the present study cannot make causal inferences. Ideally, an experimental study manipulating a green space’s vs. urbanized space’s crowdedness should be conducted in addition.
Second, as suggested in the discussion part, prosociality should be seen as multidimensional [65]. The SVO slider measure is one-dimensional and cannot account for the multitude of prosocial behaviors that exist. Therefore, it is possible that a measure that considers the multidimensionality of prosocial behaviors would produce more nuanced results.
Moreover, prosocial behaviors vary depending on the target person [66]. In the case of the present study, the results only concern prosocial behavior towards a stranger and therefore cannot be extended to a situation with friends or family members.
In addition, data do not allow us to know to what extent the perception is driven by pandemic or by pre-pandemic constructs. The need for uncrowded GS may be either a need related to the pandemic situation or a need that existed long before. Some studies comparing pre- and post-pandemic situations show that the reason for visiting GS changed during the pandemic, with a decrease in activities that could be considered as high-risk activities such as meeting people [40], and an increase in GS attendance for stress relief [16]. Considering that the main reason for the decrease of GS attendance seems to be the fear of the coronavirus (and not other reasons like the closure of the GS, governments’ incentives to stay home, or the lack of need to go out for example [16], it is possible that the perception, needs, and behaviors towards these spaces are different outside the pandemic period. It would therefore be interesting to replicate this study outside of a pandemic situation.
Last but not least, given that the sample is only composed of French-speaking people, it is possible that the obtained results are only applicable to Westernised cultures. As already mentioned, human perceptions and preferences towards natural environments significantly differ between countries [57,58], indicating that environmental factors like COVID-19, but also cultural background can condition the perception, expectations, and behaviors toward nature [40,54]. This cultural aspect therefore potentially influences our results.”
Point 4: MINOR ISSUES - The manuscript needs to be read by a NATIVE English speaker as there are some awkward coices of words, presumably due to the fact that the authors are Francophone.
Response 4: We had a native person proofread the document and make some corrections. Due to the relatively short response time, this proofreading could not be as thorough as desired. However, the document will be submitted for proofreading to the journal's Free Language Editing Services, if accepted for publication.
